# Assessment of the Antibiofilm Performance of Chitosan-Based Surfaces in Marine Environments

**DOI:** 10.3390/ijms232314647

**Published:** 2022-11-24

**Authors:** Marta Lima, Luciana C. Gomes, Rita Teixeira-Santos, Maria J. Romeu, Jesus Valcarcel, José Antonio Vázquez, Miguel A. Cerqueira, Lorenzo Pastrana, Ana I. Bourbon, Ed D. de Jong, Jelmer Sjollema, Filipe J. Mergulhão

**Affiliations:** 1LEPABE—Laboratory for Process Engineering, Environment, Biotechnology and Energy, Faculty of Engineering, University of Porto, Rua Dr. Roberto Frias, 4200-465 Porto, Portugal; 2ALiCE—Associate Laboratory in Chemical Engineering, Faculty of Engineering, University of Porto, Rua Dr. Roberto Frias, 4200-465 Porto, Portugal; 3Grupo de Reciclado y Valorización de Materiales Residuales (REVAL), Instituto de Investigaciones Marinas (IIM-CSIC), C/Eduardo Cabello, 6, CP36208 Vigo, Spain; 4International Iberian Nanotechnology Laboratory, Department of Life Sciences, Av. Mestre José Veiga s/n, 4715-330 Braga, Portugal; 5Department of Biomedical Engineering, University Medical Center Groningen, University of Groningen, Antonius Deusinglaan 1, 9713 AV Groningen, The Netherlands

**Keywords:** chitosan, marine biofouling, marine waste, biofilm formation, antifouling coatings

## Abstract

Marine biofouling is a natural process often associated with biofilm formation on submerged surfaces, creating a massive economic and ecological burden. Although several antifouling paints have been used to prevent biofouling, growing ecological concerns emphasize the need to develop new and environmentally friendly antifouling approaches such as bio-based coatings. Chitosan (CS) is a natural polymer that has been widely used due to its outstanding biological properties, including non-toxicity and antimicrobial activity. This work aims to produce and characterize poly (lactic acid) (PLA)-CS surfaces with CS of different molecular weight (Mw) at different concentrations for application in marine paints. *Loligo opalescens* pens, a waste from the fishery industry, were used as a CS source. The antimicrobial activity of the CS and CS-functionalized surfaces was assessed against *Cobetia marina*, a model proteobacterium for marine biofouling. Results demonstrate that CS targets the bacterial cell membrane, and PLA-CS surfaces were able to reduce the number of culturable cells up to 68% compared to control, with this activity dependent on CS Mw. The antifouling performance was corroborated by Optical Coherence Tomography since PLA-CS surfaces reduced the biofilm thickness by up to 36%, as well as the percentage and size of biofilm empty spaces. Overall, CS coatings showed to be a promising approach to reducing biofouling in marine environments mimicked in this work, contributing to the valorization of fishing waste and encouraging further research on this topic.

## 1. Introduction

Marine biofouling is a spontaneous and complex process by which natural and artificial submerged structures are colonized by marine organisms [1]. This undesirable attachment of molecules and fouling organisms has been recognized as a concern in the marine industry since it is responsible for several economic, industrial, environmental, and health-related implications [2,3]. The presence of organisms on marine vessels increases the weight of ships and their drag resistance, resulting in higher fuel consumption and environmental pollution [4]. Moreover, biofouling changes the physicochemical properties of marine surfaces, promoting their fast deterioration and corrosion, and can also contribute to species invasion, causing negative effects on global biodiversity [5]. Marine biofouling can also affect partially submerged equipment used for monitoring dissolved oxygen, turbidity, and pH, resulting in incorrect measurements [6]. All these consequences create a massive economic and ecological burden, stressing the need to develop new approaches to protect submerged surfaces from biofouling organisms.

Biofouling in the marine environment is a dynamic process that usually involves three steps: conditioning film formation, microfouling, and macrofouling [7]. Conditioning film is formed by the adsorption of organic molecules on submerged surfaces and promotes microfoulers (e.g., bacteria and diatoms) adhesion and consequently biofilm formation (microfouling). Biofilms established on surfaces promote the settlement of macrofoulers (such as sponges, mussels, and algae) and, within days to weeks, macrofouling communities are completely established over the submerged surfaces [8]. Since biofilm formation is one of the first steps of this natural process, a potential strategy to delay macrofouling is to prevent adhesion and biofilm formation by marine bacteria, which are early marine surface colonizers. To date, several antifouling paints have been used to prevent biofouling on ship hulls, mainly by the gradual erosion and release of biocides and toxic chemicals [9]. However, as these antifouling agents can persist in the environment and pose a threat to marine organisms, the International Maritime Organization (IMO) has banned their use in the production of antifouling paints [10,11]. Therefore, the development of novel, non-toxic and eco-friendly approaches to prevent marine biofouling in ship hulls, such as bio-based coatings, is urgently required.

Among different biopolymers, chitosan (CS) has received significant attention from academia and industry for its many applications. CS is a cationic polysaccharide obtained by the deacetylation of chitin, which is the second most abundant polymer on Earth and is commonly sourced from crustacean shells, mollusks, insects, and fungi [12,13]. The use of chitin and CS can be advantageous in solving some environmental problems, and in the last few decades, squid pens have been increasingly explored as a source of chitin. Since the average yield of edible flesh in squid is around 70% [14], squid processing produces a substantial amount of waste that ranges from 0.8 to 1.6 million tonnes per year [15]. Therefore, to avoid the costly disposal of this waste and enhance the potential of chitin and CS valorization, integration into a biorefinery and a circular economy strategy were suggested. These aim to benefit both the economy and the environment through the sustainable conversion of chitin and CS into nitrogen-rich chemicals for various applications (e.g., pharmaceuticals, cosmetics, and water treatment) [16].

Besides the use of CS enabling the valorization of fish processing industry discards, CS has been widely used due to its interesting intrinsic properties including non-toxicity, biocompatibility, film-forming ability, chemical stability, low cost, and antimicrobial activity against a broad spectrum of microorganisms [17]. Although the CS mechanism of action is not entirely known, three main mechanisms have been proposed for the inhibition of microbial growth: (i) cell membrane disruption, as a result of electrostatic interactions between the positively charged CS molecules and the negatively charged cell membranes, which can lead to loss of intracellular content and cell death [18,19]; (ii) inhibition of protein synthesis that can occur when CS molecules penetrate microbial cells, complex with DNA and inhibit mRNA synthesis [20]; and (iii) chelation of CS molecules with some metals ions, which damages the microorganism cell wall [18,21,22].

Although the antifouling activity of CS-based coatings has been reported by our group for other applications (food packaging and medical settings) [23,24] and short-term applications in the marine field [25], in vitro studies to test the long-term performance of CS coatings under operational conditions that simulate marine environments remain scarce. In addition, since the antimicrobial activity of CS and its derivatives depends on a set of structural properties such as molecular weight (Mw), degree of deacetylation (DD), concentration, and source [20,26], studies based on the physical and chemical properties of CS and their influence on biofilm formation are required for the development of more effective antifouling surfaces.

The present study aims to (i) produce and characterize poly (lactic acid) (PLA) surfaces coated with CS of different Mw and concentrations obtained from the *Loligo opalescens* pen, and (ii) evaluate the antifouling activity of these surfaces against *Cobetia marina* biofilm formation. Besides, the mechanism of action of this type of CS was clarified. To the best of our knowledge, this is the first study that encompasses the crucial steps for the synthesis and characterization of PLA-CS films for application on marine surfaces, with CS recovered from marine by-products. Moreover, this is the first study that reveals the potential of PLA-CS surfaces to reduce *C. marina* fouling on underwater surfaces under nutritional conditions, temperature, and hydrodynamics that mimics the conditions typically found in marine environments. *C. marina* DSMZ 4741 is a ubiquitous bacterium isolated from coastal seawater [27] and was chosen as a microfouler model [28]. Considering the goal of developing antifouling paints for ship hulls, PLA was the substrate chosen for this proof of concept since it has been used in several environmental-friendly antifouling approaches, including the production of marine coatings [29,30,31]. Furthermore, it is described that PLA does not biodegrade in normal ambient conditions or marine environments, and offers mechanical stability, with no changes in mechanical properties after submersion tests in the sea [32].

## 2. Results

### 2.1. Chitosan and Its Mechanism of Action

Chitosan was first extracted from endoskeleton by-products of *L. opalescens* squid through a combination of enzymatic and alkaline treatments, according to the conditions fully described in previous work [23,24]. A highly purified CS (β-CS) with an Mw of 294 kDa and a 92% degree of deacetylation was recovered and submitted to depolymerization through the reaction with sodium nitrite. This generated three derivatives of different Mw: CS1, CS2 and CS3 of 186, 129 and 61 kDa, respectively. CS chemical structures are represented in Appendix A.

In order to clarify the CS mechanism of action, *C. marina* cells were exposed to 0.5 and 1% β-CS for 24 h and then stained with Bis-(1,3-Dibutylbarbituric Acid) Trimethine Oxonol (DiBAC_4_(3), a membrane potential marker) and propidium iodide (PI, a membrane integrity marker) and analyzed by flow cytometry. Figure 1a,b shows the fluorescence intensity (FI) of *C. marina* after staining with DiBAC_4_(3). Bacterial cells exposed to both β-CS concentrations displayed a higher FI than non-treated cells (approximately 50-fold higher; *p* < 0.05). In addition, there were no significant differences in the FI of cells treated with 0.5 and 1% β-CS. As DiBAC_4_(3) enters only into depolarized cells, these results indicate that exposure to β-CS induces depolarization of the cell membrane.

Likewise, the effect of β-CS on cell membrane integrity was also investigated by staining cells with PI. Figure 1c,d presents the FI and the percentage of PI-positive cells, respectively. Data show that the exposure either to 0.5 or 1% β-CS increased the percentage of PI (+) cells (about 20%) compared to non-treated cells (5%; *p* < 0.05). Besides, there were no significant differences between the percentage of PI (+) cells when treated with 0.5 and 1% β-CS. Considering that PI is a fluorescent molecule that intercalates the DNA of membrane-compromised cells, there is evidence that β-CS causes cell membrane lesions.

### 2.2. Synthesis and Characterization of Poly (Lactic Acid)-Chitosan Surfaces

Solutions of β-CS and its three derivatives at 0.5% and 1% (*w*/*v*) were immobilized onto PLA films by dip coating. The produced surfaces were analyzed concerning their water contact angle and roughness. Regarding hydrophobicity, the water contact angles of the control (PLA) and the four PLA-CS surfaces (PLA-β-CS, PLA-CS1, PLA-CS2, and PLA-CS3) were measured using the sessile drop method. Figure 2a shows the effect of CS immobilization on the water contact angle of the PLA film. Considering that a hydrophobic surface is characterized by water contact angle values above 90° [33], results demonstrate that all surfaces (PLA, and 0.5% and 1% (*w*/*v*) PLA-CS surfaces) present a hydrophilic behaviour (Figure 2a). Regardless of the concentration, a significant reduction of approximately 30° in the water contact angle value was observed on CS-coated PLA compared to the non-functionalized PLA, ensuring that CS was successfully immobilized. In general, no significant differences between the water contact angle values for 0.5% and 1% CS surfaces were observed, revealing that CS concentration did not influence surface wettability. Likewise, the Mw of the immobilized CS did not affect the surface hydrophobicity since all surfaces functionalized with the chitosan derivatives (CS1, CS2, and CS3) show similar contact angle values to surfaces coated with the native chitosan (β-CS), about 40°.

To determine the surface roughness of the control and CS-based surfaces, profilometry analysis was performed. Average roughness ($S_{a}$) values (Figure 2b) between 236 and 380 nm revealed the smoothness of both PLA and functionalized surfaces. Moreover, no significant differences in PLA-CS roughness were observed for the different types of CS tested (*p* > 0.05).

### 2.3. Antifouling Activity of CS-Based Surfaces

The antifouling performance of the CS-based surfaces was evaluated against *Cobetia marina* biofilms developed for 49 days under hydrodynamic conditions that mimic the conditions typically found in some marine environments. The biofilm cell culturability was determined by Colony-Forming Unit (CFU) counts, while biofilm thickness and structure were analyzed by Optical Coherence Tomography (OCT).

The values of culturable cells of *C. marina* biofilms formed on 0.5% and 1% (*w*/*v*) CS-based surfaces after 49 days of incubation are presented in Figure 3a,b. The analysis of biofilm composition shows that both 0.5% and 1% (*w*/*v*) CS surfaces significantly reduced the number of culturable cells compared to PLA surfaces, except for the surface coated with the native CS (PLA-β-CS). Indeed, the *C. marina* biofilms developed on 0.5% (*w*/*v*) PLA-CS1, -CS2, and -CS3 surfaces exhibited, on average, 44% ± 15% (0.24 log reduction), 55% ± 7% (0.34 log reduction) and 62% ± 13% (0.41 log reduction) fewer culturable cells, respectively, than PLA (*p* < 0.05, Figure 3a). Comparing the bactericidal behaviour of immobilized native CS (β-CS) with its derivatives (CS1, CS2, and CS3 with 186, 129, and 61 kDa, respectively), it was observed that the antimicrobial effect was higher on PLA films coated with depolymerized CS than on PLA films coated with the native CS (*p* < 0.05, Figure 3a). Regarding the antimicrobial efficacy of 1% (*w*/*v*) PLA-CS surfaces, a similar tendency was observed, since CS1-, CS2- and CS3-PLA significantly reduced the number of biofilm culturable cells by 56% ± 16% (0.37 log reduction), 68% ± 1% (0.51 log reduction), and 54% ± 15% (0.34 log reduction), respectively, compared to the control (*p* < 0.05, Figure 3b). Although there was an average difference of 11% in the percentages of reduction between the two concentrations studied (0.5% and 1% CS), this was not significant, which can be justified by the similarity of the physical and chemical properties of the surfaces functionalized with 0.5% and 1% (*w*/*v*) CS.

Regardless of CS concentration, the surfaces with the greatest antimicrobial activity were those coated with CS1, CS2 and CS3 with 186, 129 and 61 kDa, respectively (Figure 3a,b). Since these surfaces presented a significantly lower number of culturable cells compared to the control (PLA) and the native CS (β-CS), the depolymerization of CS extracted from *L. opalescens* pens enhanced its antimicrobial performance against *C. marina* under the experimental conditions used in this study.

*C. marina* biofilm thickness was analyzed by OCT imaging and the results are presented in Figure 3c,d. Regarding 0.5% CS surfaces (Figure 3c), a significant reduction was observed for all the functionalized surfaces compared to PLA. This reduction was more noticeable on biofilms developed on PLA-CS1, PLA-CS2, and PLA-CS3 surfaces (on average, 32% ± 2% less compared to the control). Likewise, for 1% CS surfaces (Figure 3d), the highest values were observed on PLA and the incorporation of CS on PLA films significantly reduced the thickness of *C. marina* biofilms. Once again, the lowest values of biofilm thickness were observed on the PLA-CS surfaces with lower Mw CS (CS2 and CS3), achieving reductions of up to 28% ± 2%. In general, no significant differences between the concentrations of 0.5% and 1% CS were observed for biofilm culturability and thickness.

This study of the effect of CS coatings on biofilm development was complemented by the analysis of the architecture of *C. marina* biofilms by OCT. Figure 4 shows representative three-dimensional (3D) OCT images. Regardless of CS concentration, biofilms developed on PLA-CS surfaces presented visible differences in their structures compared to those grown on PLA films. While biofilms formed on the functionalized surfaces were more homogeneous and compact (Figure 4b–i), those formed on the control surface presented more prominent and irregular structures (Figure 4a). Moreover, no considerable differences between the 3D structures of biofilms formed on 0.5% (Figure 4b–e) and 1% (*w*/*v*) CS surfaces (Figure 4f–i) were observed. All these results are corroborated by the biofilm cell counts and thickness, revealing that both native CS and its derivates prevent *C. marina* biofilm development.

Since biofilm growth can be influenced by the spatial distribution of microorganisms [34], *C. marina* biofilms were also analyzed regarding the percentage and size of empty spaces. Figure 5 shows the quantitative data and Figure 6 presents a 2D graphical representation of biofilm empty spaces on each surface. The mean percentage of empty spaces (Figure 5a,b) ranged from 1.5% ± 0.4% to 4.3% ± 0.3% for biofilms developed on 1% (*w*/*v*) PLA-CS2 and PLA surfaces, respectively. Regarding the size of empty spaces obtained for *C. marina* biofilms formed on the different surfaces, mean values are presented in Figure 5c,d, and range from 1487 µm^2^ to 5454 µm^2^ for biofilms developed on 0.5% (*w*/*v*) PLA-CS3 and PLA surfaces, respectively. Regardless of chitosan concentration, the highest values of both parameters were obtained for biofilms developed on PLA films, while the lowest values were detected on PLA-CS surfaces. Although, in general, there were no considerable differences in the percentage and size of empty spaces of biofilms formed on the CS-based surfaces, the biofilms developed on PLA surfaces coated with depolymerized CS (PLA-CS1, -CS2, and -CS3) show significantly lower percentages and size of empty spaces compared to PLA and PLA coated with the native CS (PLA-β-CS).

## 3. Discussion

Given the economic and ecological effects of marine biofouling, the development of antifouling strategies for marine environments is imperative. Although some antifouling paints, such as biocide-containing paints, have been used to reduce the propensity of biofouling, the rigid international regulations and environmental concerns call for sustainable and environmentally friendly antifouling approaches, such as bio-based coatings [35,36]. In this study, CS-based surfaces with different concentrations and Mw were produced and characterized, and their long-term performance in preventing biofilm formation of *C. marina* was evaluated through an analysis of biofilm cell amount and architecture. In order to increase the predictive value of this work, this analysis was performed under laboratory conditions that mimic real marine environments. *C. marina* biofilms were developed at 25 °C for 49 days [37] under a shear rate of 40 s^−1^, close to the shear rate reported for a ship hull anchored in a port (50 s^−1^) [38]. In a previous work of the group, by testing an innovative multifunctional coating [39], this methodology was shown to provide similar results when compared to surface immersion in a real marine environment for 2.5 years.

The first part of this study consists of extracting chitin from by-products of the fishery industry and producing native CS (Mw of 294 kDa) and its derivates with different Mw (CS1, CS2, and CS3 with 186, 129, and 61 kDa, respectively). Although CS is commercially available, its extraction from *L. opalescens* pens enables the valorization of the fish processing industry discards and is an economically and environmentally sustainable strategy. This species of squid, together with *Illex argentinus*, is the most captured around the world [40,41]. To the best of our knowledge, this is the first work dealing with the application of β-chitosan isolated from squid pen against marine bacteria. Moreover, most of the chitin commercially available is in the form of α-chitin, which can be extracted from crustaceans shells and is characterized by its antiparallel polymeric chains [12]. Conversely, β-chitin isolated from squid pens has parallel polymer chains connected by hydrogen bonds which, due to its alignment, create inter- and intra-molecular forces weaker than those found in α-chitin, increasing its water-absorbing capacity and its solubility [40].

Considering that β-chitosan isolated from squid pen was tested against *C. marina* for the first time, we sought to clarify its mechanism of action. Flow cytometric analysis indicates that β-CS targets the bacterial cell membrane inducing its depolarization and pore formation. Since the β-CS effect was more measurable at the membrane potential level than at the membrane integrity level after 24 h exposure, these two events likely occur in cascade. Moreover, the β-CS mode of action was independent of the tested concentrations. In fact, several authors have postulated that CS disrupts cell membranes as a result of electrostatic interactions between the positively charged CS molecules and the negatively charged cell membranes, leading to loss of intracellular content and cell death [18,19,22,23,24,42].

After the functionalization of the PLA-CS surfaces by dip coating, they were characterized regarding their wettability and roughness as these properties can affect their antimicrobial activity [43,44]. Water contact angles reveal that all surfaces exhibited a hydrophilic behaviour, and the CS immobilization decreased the water contact values. Similarly, previous studies demonstrate that the incorporation of CS molecules increased the hydrophilicity of the surfaces due to the hydrophilic nature of the polymer and the increase of polar groups in the coatings [45,46,47,48]. Since bacterial adhesion is favored by the hydrophobic character of surfaces, the immobilization of CS on PLA films may reduce microbial attachment and subsequent biofilm formation [49]. Additionally, the wettability of the PLA-CS surfaces was not dependent on CS concentration and Mw. Similar results were obtained by Ururahy et al. [50], which show that different CS concentrations did not influence the wettability of the substrate. Moreover, Stoleru et al. [51] revealed that the wettability of PLA films functionalized with different CS was not affected by CS Mw. Concerning profilometry analysis, the PLA and CS-based surfaces display similar roughness values, regardless of the CS concentration and Mw. Although some studies have reported that the surface roughness increases with the deposition of CS [52,53], this effect is highly dependent on CS properties (Mw and degree of deacetylation) and the coating method [51]. Overall, although the characterization of the PLA-CS surfaces indicates that CS was successfully incorporated onto PLA films, the effect of CS Mw and concentration on the surface properties was not significant.

Lately, several researchers have studied the antifouling performance of CS combined with other compounds, such as zinc oxide and copper oxide, to improve its antifouling properties and stability [17,54,55]. To the best of our knowledge, this is the first study that evaluates the antifouling performance of PLA surfaces coated with native CS and its derivatives without adding any other compound against *C. marina*.

Results from biofilm cell culturability indicate that *C. marina* biofilms formed on 0.5% and 1% (*w*/*v*) CS-based surfaces presented a significantly lower number of culturable cells compared to those grown on PLA films, revealing the antimicrobial performance of the functionalized surfaces. Although the efficacy of an antifouling coating may be dependent on a wide range of environmental factors, such as salinity, availability of nutrients, hydrodynamics, and organisms [7,56], the results obtained are supported by the literature. The antimicrobial activity of CS-based coatings with CS concentrations below 2% (*w*/*v*) has already been reported against some fouling microorganisms, including *Bacillus* sp., *Pseudomonas* sp., and *Vibrio* [57,58,59]. Jena et al. [60] investigated the effect of CS-based coatings on biofilm density and revealed that coated surfaces allowed to reduce *Pseudomonas* sp. density (CFU·cm^−2^) by 84% compared to uncoated specimens. Moreover, previous laboratory and mesocosm experiments performed by Al-Naamani et al. [54] revealed that CS paints were able to significantly reduce the density of diatom *Navicula incerta* and marine fouling bacteria *Pseudoalteromonas nigrifaciens*. Elshaarawy et al. [61] evaluated the antibacterial effect of CS against a range of significant biofilm-inducing bacterial strains such as *Escherichia coli*, *Staphylococcus aureus*, *Aeromonas hydrophila* and *Vibrio*, revealing that CS presents a higher bactericidal activity compared to a standard antifoulant Diuron. Moreover, Al-Naamani et al. [62] show that plastic films coated with 2.5% CS significantly reduced the settlement of *Bugula neritina* compared to uncoated plastic films. After being incorporated into a marine paint and applied to plastic substrates, CS-based coatings were found to inhibit bacterial fouling over one week, and to significantly reduce the cell density of fouling bacteria after two weeks of immersion in a natural seawater environment. These results were corroborated by Dobretsov et al. [63], who disclosed that CS paints significantly reduced the biofouling on surfaces exposed to the environmental conditions in the Sea of Oman, emphasizing the potential of CS to be applied in protective paints.

The antimicrobial activity of CS-based coatings is dependent on a range of intrinsic and extrinsic factors, including microorganism species, surface wettability and roughness, CS degree of deacetylation, concentration, and Mw [20,26]. In the present work, surfaces with different CS concentrations and Mw were produced and show different antifouling performances. Comparing the results obtained for 0.5% and 1% (*w*/*v*) CS surfaces, no significant differences in the number of *C. marina* biofilm culturable cells were observed, corroborating a previous study published by Al-Belushi et al. [64], where the effect of 1% and 2% (*w*/*v*) CS coatings against a Gram-negative bacteria was assessed. Since all CS-based surfaces presented similar values of water contact angles and roughness, it is not expected that the physicochemical properties and morphology of the surfaces directly impact their antimicrobial activity. In addition, flow cytometric experiments indicate that 0.5 and 1% β-CS treatments yield a similar antimicrobial effect. Therefore, CS Mw seems to be the main parameter to influence the bactericidal performance of the functionalized surfaces. Indeed, regardless of CS concentration, the PLA-CS surfaces show different bactericidal performances; the reduction of *C. marina* culturability is higher on the PLA films coated with CS of lower Mw. There is no general agreement on the relationship between the CS Mw and its antimicrobial activity. No et al. [59] demonstrated that oligo-CS with an Mw of 1–10 kDa had a lower antimicrobial activity compared to CS of higher Mw (22–1671 kDa). Moreover, no significant differences were found between the antibacterial performance of low (22 and 59 kDa), medium (224 kDa), and high (470, 746, 1106, and 1671 kDa) CS Mw [59]. The effect of CS Mw (2–16 kDa) was also assessed in a study developed by Simunek et al. [65] which revealed that CS antimicrobial activity increased with the increase of CS Mw. On the other hand, some studies have shown that lower Mw CS is more effective. Tayel et al. [66] evaluated the effect of Mw on the antimicrobial activity of CS with 21, 27, 140, and 190 kDa and showed that, in general, decreasing the Mw of CS slightly increased its antimicrobial activity. Likewise, Benhabiles et al. [67] demonstrated that a native CS extracted from shrimp shell waste presented a reduced antimicrobial activity compared to its derivates of lower Mw. Moreover, Zheng et al. [68] revealed that the antimicrobial performance of CS was higher with lower Mw against Gram-negative bacteria, but not against Gram-positive bacteria, which corroborates the results obtained in the present study.

Since the biofilm structure can impact its resistance to mechanical and chemical agents, such as fluid shear and antifouling compounds [69], the effect of immobilized CS and its derivates against biofilm formation was also analyzed by OCT imaging. Both quantitative data of biofilm thickness and 3D biofilm structures highlighted the effect of CS on *C. marina* biofilm growth, demonstrating that all functionalized surfaces had thinner and more compact biofilms than PLA films. A less compact structure combined with the presence of streamers on biofilms developed on PLA surfaces can enhance biofilm growth by promoting the transfer of nutrients to the inner layers and the capture of new cells and other components to the biofilm [70]. These results corroborate the biofilm culturable cell analysis and are in accordance with previous studies where the effect of CS-based paints on biofilm structure was evaluated. El-Saied et al. [71] investigated the antifouling performance of a CS-based marine paint by immersing coated PVC panels in the Mediterranean Sea Eastern Harbor of Alexandria. The findings revealed the long-term antifouling activity of CS since the coatings inhibited the development of tubeworms and barnacles on panels submerged for more than two months. Similarly, Elshaarawy et al. [72] showed that CS-coated panels were highly efficient against tube worms, barnacles, and macroalgae settlement, even when compared to a standard antifoulant.

Within the biofilm, microorganisms are commonly organized in specialized niches, where heterogeneous microenvironments exist as a result of nutrient transport and chemical gradients [34]. Quorum sensing, intracellular communication, and consequently biofilm formation can be affected by the spatial distribution of microorganisms, which is impacted by microenvironmental factors [69]. The presence of empty spaces in biofilms can impact the structure of microbial communities and consequently their diversity, activity, and synergism [34,73]. In the present work, OCT analysis revealed that PLA-CS surfaces show lower percentage and mean size values of empty spaces compared to uncoated PLA films. The empty spaces could be beneficial for mass transport within the biofilm, enhancing the distribution of nutrients and oxygen and providing a way for the removal of metabolic end-products [74]. Thus, biofilms with a higher percentage and larger size of empty spaces, such as the biofilms developed on PLA surfaces, are more prone to the flow of medium throughout the biofilm, which may result in the establishment and expansion of a channel network, relieving nutrient limitations and promoting biofilm growth [74,75].

Altogether, these findings suggest that PLA coated with CS extracted from fishery industry discards may provide an efficient and environmentally friendly approach to retard biofouling in submerged surfaces. Moreover, the present work reveals that the effect of CS Mw strongly affects biofilm cell number and architecture, and the underlying mechanism of the antimicrobial effect of studied CS was cell membrane depolarization and consequent integrity loss.

## 4. Materials and Methods

### 4.1. Chitosan and Its Derivates

Chitosan was extracted from the *Loligo opalescens* squid endoskeletons (pens) through a combination of enzymatic and alkaline treatments [41]. Briefly, after being milled, squid pens were deproteinized by a protease (alcalase from Novozymes, Bagsvaerd, Denmark) to generate chitin, which was converted into chitosan through a NaOH treatment. Subsequently, the obtained CS (β-CS) was submitted to a depolymerization process using sodium nitrite [76], generating three different depolymerized CS (CS1, CS2, and CS3). All samples were freeze-dried and milled to powder. Nuclear magnetic resonance spectroscopy and gel permeation chromatography were used to determine the native CS’s degree of deacetylation, and the molecular weights of β-CS and its depolymerized derivatives, respectively.

### 4.2. Bacteria Strains and Culture Conditions

The CS mechanism of action and antifouling properties of the functionalized surfaces were evaluated against *Cobetia marina* DSMZ 4741 (obtained from the German Culture Collection, DSMZ, Braunschweig, Germany), a ubiquitous bacterium isolated from aquatic ecosystems that is known to form biofilms [27,77]. Stock cultures were preserved at −80 °C in the complex marine medium Våatanen Nine Salt Solution (VNSS) with 20% (*v/v*) glycerol. The VNSS medium was prepared as previously described [78] and used to simulate the nutritional conditions found in marine environments. Before each experiment, bacteria were spread on VNSS supplemented with 15 g·L^−1^ agar (VWR International, Leuven, Belgium) and incubated at 25 °C for 24 h. The starting cultures were prepared by collecting single colonies from VNSS agar plates to 250 mL of VNSS broth and incubating at 25 °C, 160 rpm for 16 h ± 2 h. The overnight cultures were centrifuged (Eppendorf Centrifuge 5810R, Eppendorf, Hamburg, Germany) at 18 °C, 3772× *g* for 10 min, resuspended in fresh VNSS medium, and adjusted to a final suspension of 1 × 10^8^ CFU·mL^−1^.

### 4.3. Characterization of CS Mechanism of Action

The mechanism of action of CS directly obtained from the *L. opalescens* pens was characterized in *C. marina* by flow cytometry. Briefly, a bacterial suspension was treated with 0.5 and 1% of β-CS at 25 °C for 24 h. Subsequently, cells were harvested by centrifugation at 9391 *g* (Eppendorf Centrifuge 5424, Eppendorf, Hamburg, Germany) for 10 min at room temperature and resuspended in 8.5 g·L^−1^ sterile sodium chloride (NaCl) solution (VWR International, Carnaxide, Portugal). The evaluation of cell membrane potential was performed by staining cells with Bis–(1,3—Dibutylbarbituric Acid) Trimethine Oxonol (DiBAC_4_(3), Sigma–Aldrich, Taufkirchen, Germany) at 0.5 µg·mL^−1^ for 30 min at 25 °C, in absence of light. DiBAC_4_(3) is a membrane potential-sensitive dye that enters only depolarized cells, where it binds reversibly to intracellular proteins or membranes, resulting in an increased fluorescent signal [79]. In turn, cell membrane integrity was assessed by staining cells with propidium iodide (PI, Invitrogen Life Technologies, Alfagene, Lisboa, Portugal) at 1 µg·mL^−1^ for 30 min at 25 °C in the dark. PI is an indicator of membrane damage as it is a red fluorescent double-charged cationic molecule that intercalates double-stranded DNA of compromised cells, but that usually does not penetrate intact membranes [80]. After staining, cells were analyzed in a CytoFLEX flow cytometer model V0-B3-R1 (Beckman Coulter, Brea, CA, USA), using the CytExpert software (version 2.4.0.28, Beckman Coulter, Brea, CA, USA). Samples were acquired at a flow rate of 10 µL·min^−1^. The fluorescence intensity at FL1 (fluorescent detector; 530 nm) was registered for DiBAC_4_(3), while the percentage of PI-positive (PI (+)) cells at FL3 (fluorescent detector; 610 nm) was recorded for PI. All assays were performed in duplicate.

### 4.4. Functionalization of Poly (Lactic Acid)-Chitosan Surfaces

Solutions of 0.5% and 1% (*w*/*v*) of β-CS and its three derivatives were immobilized onto poly (lactic acid) (PLA) films (Goodfellow, Cambridge, UK) through dip coating. PLA was the substrate chosen because it has been used in eco-friendly antifouling strategies, including the production of marine coatings [29,30,31]. First, to improve the CS adhesion to the surfaces, PLA films were submitted to plasma oxygen treatment (Harrick Plasma, Ithaca, NY, USA) for 15 min [81]. Subsequently, PLA films (1 × 1 cm) were dipped in the different CS solutions (0.5% and 1% (*w*/*v*) β-CS, CS1, CS2, and CS3) for 15 min and dried with nitrogen for 5 min [81].

### 4.5. Surface Characterization

#### 4.5.1. Water Contact Angle Measurements

The surface hydrophobicity was determined through the measurement of water contact angles by the sessile drop method using a contact angle meter (DSA 100E, Kruss Gmbh, Hamburg, Germany). Briefly, a 2 µL water droplet was placed in different positions on the surfaces, and drop images were acquired using a camera connected to the analyzer. The circle-fitting method was used to determine the water contact angles [82]. Measurements were performed at room temperature (25 °C ± 2 °C) and at least ten determinations for each surface were made.

#### 4.5.2. White Light Profilometry

White light profilometry was used to determine the average roughness ($S_{a}$) of the PLA and PLA-CS (PLA-β-CS, PLA-CS1, PLA-CS2, and PLA-CS3) surfaces, as previously performed by Whitehead et al. [83,84]. A MicroXAM surface mapping microscope (ADE corporation, XYZ model 4400 mL system) with an objective of 50× and connected to an AD phase shift controller (Omniscan, Wrexham, UK) was used to image at least three different zones of three independent assays. The images were analyzed by Mountains^®^ 9 software (version 9.2.10042, Digital Surf, Besançon, France).

### 4.6. Antifouling Activity of Chitosan-Based Surfaces

#### 4.6.1. Biofilm Formation

Biofilm assays were performed in 12-well microtiter plates (VWR International, Carnaxide, Portugal) under controlled hydrodynamic conditions. Before each experiment, all surfaces, including the PLA (control) and the four functionalized CS surfaces, were sterilized by ultraviolet radiation for 30 min, fixed on the microplate wells using double-sided adhesive tape, and inoculated with 3 mL of the bacterial suspension. Additionally, 3 mL of VNSS medium was added to the wells containing sterilized surfaces to monitor their sterility throughout the experiments. The microplates were then incubated at 25 °C in an orbital shaker with a 25 mm orbital diameter (Agitorb 200ICP, Norconcessus, Ermesinde, Portugal) at 185 rpm. A previous study described that a shaking frequency of 185 rpm in this type of incubator corresponds to an average shear rate of 40 s^−1^ [56], close to the shear rate estimated for a ship in a harbor (50 s^−1^) [38]. Biofilm formation was monitored for 7 weeks (49 days) since this period corresponds on average to half of the minimal economically viable interval accepted for the maintenance and cleaning procedures of marine submerged surfaces [37]. During the incubation period, the culture medium was replaced twice a week.

Biofilm formation experiments were performed in three independent biological assays with three technical replicates each.

#### 4.6.2. Biofilm Quantification

Briefly, the culture medium was carefully removed from the wells and the coupons were gently washed with 8.5 g·L^−1^ sterile NaCl solution to remove loosely attached bacteria. At least three coupons of each surface were then analyzed concerning the number of culturable cells by colony-forming unit (CFU) counts, and biofilm thickness and structure by Optical Coherence Tomography (OCT).

##### Colony-Forming Unit Counts

The number of biofilm culturable cells was determined by CFU counting (CFU·cm^−2^). Biofilm cell suspensions were obtained by detaching PLA and PLA-CS surfaces from the wells, dipping in 2 mL of 8.5 g·L^−1^ NaCl solution, and vortexing for 3 min at full power (ZX4, Velp Scientifica, Usmate, Italy). Bacterial suspensions were properly diluted and plated on VNSS agar plates, which were incubated overnight at 25 °C.

##### Biofilm Thickness and Structure

On day 49, biofilm thickness and structure were analyzed by Optical Coherence Tomography (OCT) using a Thorlabs Ganymede Spectral Domain Optical Coherence Tomography system with a central wavelength of 930 nm (Thorlabs GmbH, Dachau, Germany). Images from *C. marina* biofilms developed on PLA and PLA-CS surfaces were acquired and analyzed as previously described by Romeu et al. [56]. Since biofilms are mainly composed of water [85], the established refractive index was 1.40, close to the refractive index of water (1.33). For each coupon, imaging was performed with a minimum of five fields of view to ensure the accuracy and reproducibility of the results obtained. The 2D and 3D images were analyzed using a routine developed in the Image Processing Toolbox from MATLAB 8.0 and Statistics Toolbox 8.1 (The MathWorks, Inc., Natick, MA, USA).

### 4.7. Statistical Analysis

Statistical analysis was performed using the IBM SPSS Statistics version 26 for Windows (IBM SPSS, Inc., Chicago, IL, USA). Descriptive statistics were used to calculate the mean and standard deviation (SD) for cells stained with the membrane potential dye, percentage of PI (+) cells, water contact angles and surface roughness, the number of culturable cells, biofilm thickness, and the percentage and size of biofilm empty spaces. The Kolmogorov–Smirnov and Shapiro–Wilk tests were used to verify the homogeneity of variances and normality of data. Since the variables were not normally distributed, differences in the fluorescence intensity and the percentage of stained cells, the water contact angles and roughness values, as well as the number of culturable cells, biofilm thickness, and percentage and size of empty spaces of biofilms obtained for the tested surfaces (PLA and CS-based surfaces—PLA-β-CS, PLA-CS1, PLA-CS2, and PLA-CS3) were evaluated using the nonparametric Mann-Whitney test. Statistically significant differences were considered for *p*-values < 0.05, which correspond to a confidence level of 95%. All reported data are presented as mean ± SD from at least three experiments with triplicates.

## 5. Conclusions

In this study, the long-term antimicrobial and antifouling performance of CS-based surfaces against *C. marina* biofilms under conditions that mimic some marine environments were demonstrated. Although the Mw and concentration of chitosan did not impact the characteristics of the produced surfaces, the most effective antibiofilm surfaces were those coated with the lowest Mw CS, regardless of the concentration. The antimicrobial activity of the CS studied in this work was demonstrated to be linked to cell membrane depolarization with consequent loss of membrane integrity. The results obtained in this study suggest that the incorporation of CS in marine paints may be a promising eco-friendly antifouling approach to reduce the biofilm formation on ship hulls and consequently fight biofouling in this environment.

## Figures and Tables

**Figure 1 ijms-23-14647-f001:**
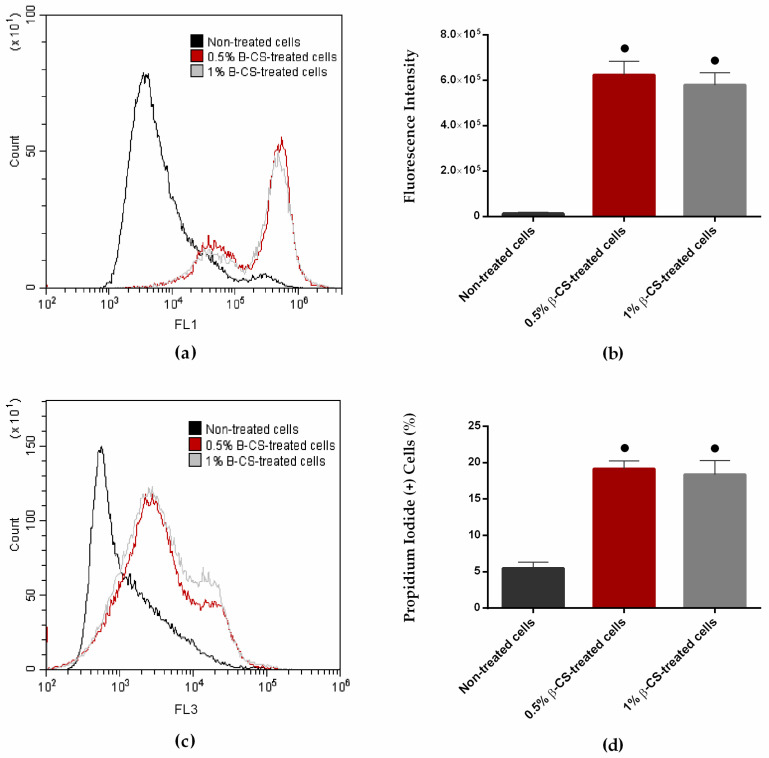
Representative flow cytometric histograms (**a**,**c**), the fluorescence intensity (**b**), and the percentage of propidium iodide-positive (PI (+)) cells (**d**) of *C. marina* non-treated (■) and treated with 0.5 (■) and 1% (■) β-CS for 24 h and stained with DiBAC_4_(3) (a membrane potential marker) (**a**,**b**) and PI (a membrane integrity marker) (**c**,**d**), respectively. The means ± standard deviations for two independent experiments are illustrated. Significant differences between treated cells and the control (non-treated cells) were evaluated using the non-parametric Mann-Whitney test and represented for *p*-values < 0.05 by ●.

**Figure 2 ijms-23-14647-f002:**
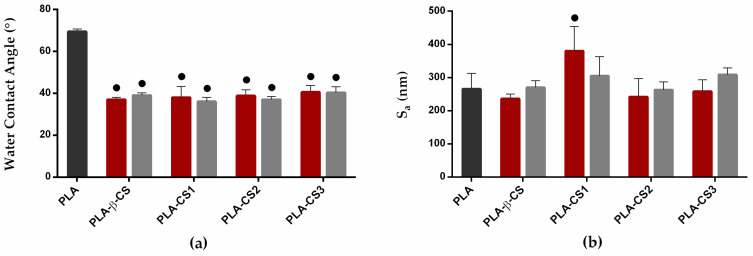
Water contact angles (**a**) and average roughness (**b**) values for the ■ PLA film, ■ 0.5% and ■ 1% PLA-CS surfaces (PLA, PLA-β-CS, PLA-CS1, PLA-CS2, and PLA-CS3). The means ± standard deviations are illustrated. Differences between PLA and the functionalized surfaces were evaluated using the non-parametric Mann–Whitney test and represented for *p*-values < 0.05 by ●.

**Figure 3 ijms-23-14647-f003:**
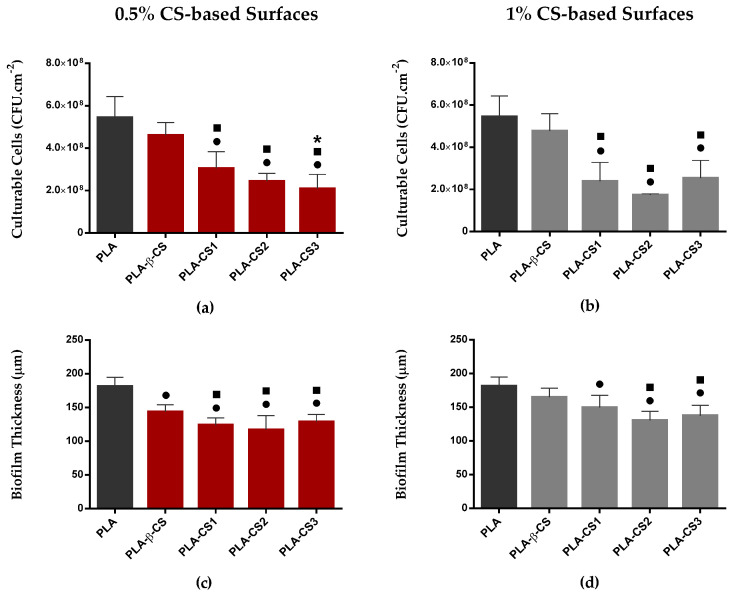
Culturable cells and thickness of *C. marina* biofilms formed on ■ PLA, and ■ 0.5% and ■ 1% (*w*/*v*) CS-coated PLA surfaces (PLA-β-CS, PLA-CS1, PLA-CS2, and PLA-CS3) after 49 days. Differences between functionalized surfaces were evaluated using the non-parametric Mann–Whitney test and represented for *p*-values < 0.05 by ●, ■, and ***** when compared to PLA, PLA-β-CS, and PLA-CS1, respectively.

**Figure 4 ijms-23-14647-f004:**
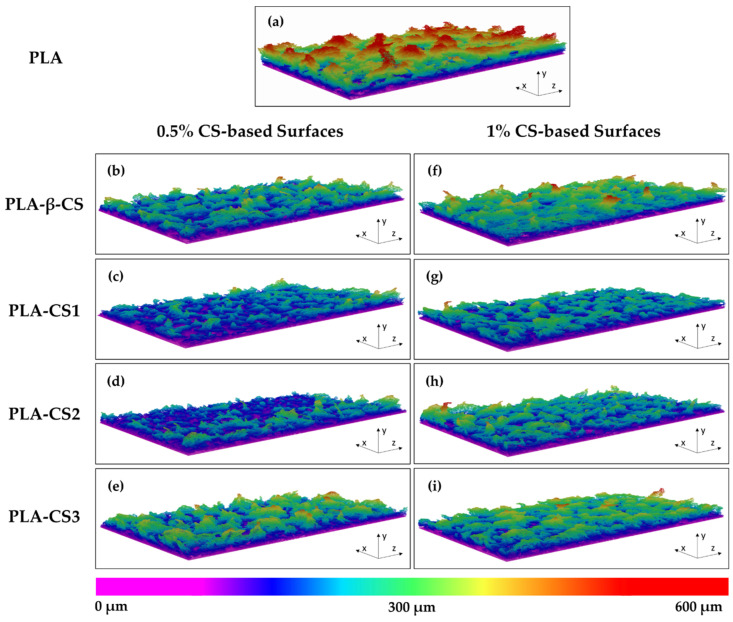
Representative 3D OCT images obtained for *C. marina* biofilms formed on PLA (**a**), and 0.5% (**b**–**e**) and 1% (*w*/*v*) (**f**–**i**) PLA-CS surfaces (PLA-β-CS, PLA-CS1, PLA-CS2, and PLA-CS3) after 49 days. The color scale shows the range of biofilm thickness. All images are represented with a scan range of 2490 µm × 1512 µm × 600 µm.

**Figure 5 ijms-23-14647-f005:**
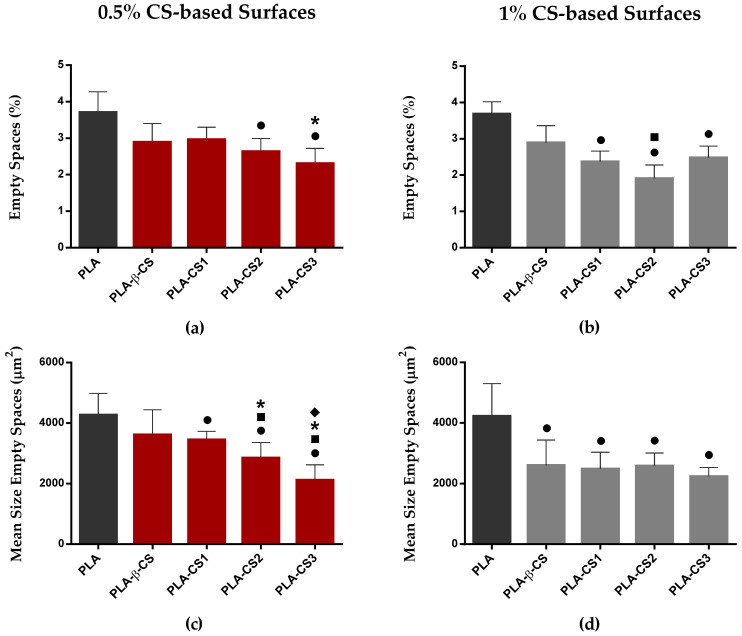
Mean percentages (**a**,**b**) and size (**c**,**d**) of empty spaces presented on ■ PLA, and ■ 0.5% and ■ 1% (*w*/*v*) CS-coated PLA surfaces (PLA-β-CS, PLA-CS1, PLA-CS2, and PLA-CS3) after 49 days. Differences between functionalized surfaces were evaluated using the non-parametric Mann–Whitney test and represented for p-values < 0.05 by ●, ■, ***** and ◆ when compared to PLA, PLA-β-CS, PLA-CS1, and PLA-CS2, respectively.

**Figure 6 ijms-23-14647-f006:**
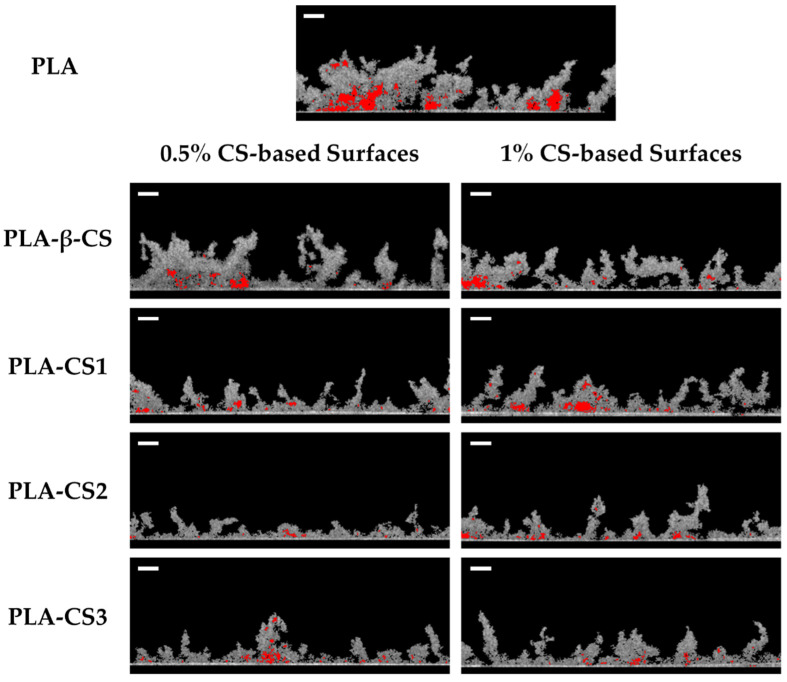
Representative 2D OCT images obtained for *C. marina* biofilms formed on PLA, and 0.5% and 1% (*w*/*v*) PLA-CS surfaces (PLA-β-CS, PLA-CS1, PLA-CS2, and PLA-CS3) after 49 days. The empty spaces of biofilms are colored in red. Scale bar = 100 µm.

## Data Availability

The data presented in this study are available on request from the corresponding author. The data are not publicly available yet as some data sets are being used for additional publications.

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
