# Peer review of "Assessment of the Antibiofilm Performance of Chitosan-Based Surfaces in Marine Environments"

_ijms, 2022, doi:10.3390/ijms232314647_

Round 1

Reviewer 1 Report

General remarks:

First of all, I would like to congratulate the authors for this work, which I really enjoyed reading. The work is very well written, presents good results, relevant to science, written clearly, concisely, well supported in the bibligraphy, basically flawless.  

The introduction is very well written. It contains information relevant to the subject researched, the objective, and the importance of the study are well defined and highlighted. Reference is made to some previous results obtained by the authors, and the justification for the current experiment is well argued in the context of the preliminary results. 

The experimental results are clearly presented and are easily perceptible.

The research methodology is clear, and duly justified using reference articles. Just one remark below.

The discussion of the results has been well elaborated and is supported by the introduction which defines the purpose of the research. Logical and coherent assumptions are issued, and the directions that need to be further researched are highlighted to elucidate issues that are not yet fully clarified.

All in all, the paper is of excelent quality and should be published with minor revision. 

Abstract:

I suggest adding information regarding Cobetia marina in the abstract: “a model proteobacterium for marine biofouling”. 

Keywords

Nothing in the abstract indicates that the paper addresses “marine waste”, so either the authors change the abstract or remove this keyword.

Introduction: 

One small remark: In Line 78. Besides a biorefinery strategy, a circular economy strategy may also be considered, for waste is being converted into products? Now I understand why there is a keyword for “marine waste”.

Results:

The results obtained are clear and remarkable: the treatment with CS clearly proved to compromise the C. marina membranes. Also, the treatments significantly decreased the formation of C. marina (numer ofcells,  biofilm thickness, size of empty spaces). 

Although most authors don’t state this, I believe that the statistical analysis used should be stated in the paper. That is, whenever the normality of the test was not achieved and the nonparametric test were used instead, that should be mentioned. It’s just a suggestion. 

Material and methods:

4.1. Chitosan and its Derivates

As a biologist I always struggle with these depolymerized structures, that are very abstract, unless I can see the actual chemical formulas. I wonder if it is possible to include the depolymerizes CS structures obtained. 

Line 427. Although the methodology has already been fully described in previous works, as this is an independent paper, the methodology should be briefly described.

Reviewer 2 Report

The paper reports on the effect of chitosan-poly(lactic acid)coating on the marine bacterium Cobetia marina. Despite the well-known antibiofouling effects of chitosan-based formulations, the authors obtained new interesting results on the physicochemical properties of surfaces covered with purified chitosan derivatives produced from fishery waste, namely squid pan as well as their antibiofilm effects. The manuscript is scientifically sound and easy to follow. I have only a few comments on improving the manuscript.

Abstract. It is worth noting the mention of the source of chitosan, especially in support of the last sentence “contributing to the valorization of marine discards”, which can be changed to “contributing to the valorization of fishing waste".

L378 On the other hand, some studies have shown that lower Mw is more effective – suggested: that lower Mw CS is more effective.
